# Fractional response analysis reveals logarithmic cytokine responses in cellular populations

Karol Nienałtowski[1], Rachel E. Rigby [2], Jarosław Walczak[1], Karolina E. Zakrzewska[1], Edyta Głów[1], Jan Rehwinkel [2] & Michał Komorowski [1✉]

Although we can now measure single-cell signaling responses with multivariate, high-throughput techniques our ability to interpret such measurements is still limited. Even interpretation of dose–response based on single-cell data is not straightforward: signaling responses can differ significantly between cells, encompass multiple signaling effectors, and have dynamic character. Here, we use probabilistic modeling and information-theory to introduce fractional response analysis (FRA), which quantifies changes in fractions of cells with given response levels. FRA can be universally performed for heterogeneous, multivariate, and dynamic measurements and, as we demonstrate, quantifies otherwise hidden patterns in single-cell data. In particular, we show that fractional responses to type I interferon in human peripheral blood mononuclear cells are very similar across different cell types, despite significant differences in mean or median responses and degrees of cell-to-cell heterogeneity. Further, we demonstrate that fractional responses to cytokines scale linearly with the log of the cytokine dose, which uncovers that heterogeneous cellular populations are sensitive to fold-changes in the dose, as opposed to additive changes.

[1] Institute of Fundamental Technological Research, Polish Academy of Sciences, Warsaw, Poland. [2] Medical Research Council Human Immunology Unit, Medical Research Council Weatherall Institute of Molecular Medicine, Radcliffe Department of Medicine, University of Oxford, Oxford, UK. ✉email: m.komorowski@sysbiosig.org

Many studies of signaling systems involve examining how the intensity of a stimulus, e.g., cytokine dose, translates into the activity of signaling effectors, e.g., transcription factors[1–7]. This is usually done by exposing cells to a range of doses and measuring responses either in bulk or at the single-cell level. Results of such experiments are then represented and interpreted in terms of dose–response curves. The standard dose–response curve depicts how the mean, median, or a characteristic of choice, changes with the increasing dose, and provides a basic, first-order model of how a signaling system operates. Several aspects of cellular signaling are difficult to analyze with mean/median dose–responses. For example, signaling responses can differ significantly between cells, encompass multiple signaling effectors, and are dynamic. First, outwardly very similar cells exposed to the same stimulus exhibit substantial cell-to-cell heterogeneity[8–11] (see refs. [12–14] for review). Therefore, the same mean/median response can result from a small fraction of strongly responding cells or a significant fraction of weakly responding cells[1,2,15]. Second, the highly interconnected architecture typical for mammalian signaling usually results in a single stimulus activating several primary signaling effectors or downstream genes[16–21]. For example, effectors of type I interferons include six members of the signal transducer and activator of transcription (STAT) family[22], which are activated with different sensitivities at different doses. Therefore, the description of dose–response in terms of an individual signaling effector is incomplete[23]. Third, live-cell imaging experiments demonstrated that the dose may not only alter the response at a single time-point but can control temporal profiles of signaling responses[24,25]. For instance, low doses of tumor necrosis factor-alpha (TNF-α) may induce one peak of nuclear factor-κB signaling activity, whereas higher doses may induce additional peaks[7,26]. Besides, the dose may control the onset, shut off, amplitude, or, in principle, any other characteristics of the responses[27–30]. Overall, mean/median dose–response curves do not capture the inherent complexity of single-cell high-throughput data, and an alternative approach is required. We have used probabilistic modeling and information-theory to develop a different analytic framework, fractional response analysis, involving fractional cell counting, which is capable of deconvoluting behaviors of heterogeneous cellular populations.

## Results

**Conventional dose–response analysis does not capture complex data.** To demonstrate the need and utility of FRA, we studied type I interferon signaling in human peripheral blood mononuclear cells (PBMCs), a system involving multiple signaling effectors, cell-to-cell heterogeneity, and several cell types. Dose–responses to the type I interferon variant IFN-α2a were analyzed via whole-cell tyrosine phosphorylation levels of effector proteins STAT1, STAT3, STAT4, STAT5, and STAT6 (pSTATs) measured jointly in individual cells using mass cytometry (CyTOF). Cells were collected from a healthy donor, and measurements were performed 15 min after IFN-α2a stimulation, the time of maximal response (Supplementary Fig. 1). Along with signaling effectors, 26 phenotypic markers (Supplementary Table 1), such as CD3 that marks T cells, were measured to allow for identification of several cell types including B cells, CD4+ T cells, CD8+ T cells, natural killer (NK) cells, and CD14+ monocytes[31–33] (Supplementary Fig. 2). Such multivariate data are often analyzed using t-SNE plots to visualize multiple cell types and signaling effectors[32,33] (Fig. 1a, b, Supplementary Fig. 3), which is a prerequisite for a more-detailed quantitative analysis usually involving mean/median dose–responses and population response distributions of individual signaling effectors. Following this

strategy, mean levels and distributions of pSTATs in B cells, CD4+ T cells, CD8+ T cells, NK cells, and CD14+ monocytes were calculated (Fig. 1c, d) and revealed that each STAT reached different maximal phosphorylation level for different doses in a particular cell type. Medians and means of the log-data (Supplementary Fig. 4a, b) yielded similar conclusions. Plotting distributions of individual signaling effectors (Fig. 1d, Supplementary Fig. 4c) exposed considerable differences in terms of cell-to-cell heterogeneity between cell types and STATs. Nonetheless, no pattern in the functioning of the signaling system was apparent. However, the data involved five signaling effectors measured in single cells of five different types resulting in a tangible complexity possibly covering any existent regularities, which highlights the need for comprehensive approaches capable of handling complex data.

**Fractional response curves.** Outcomes of physiological processes, e.g., of inflammation or stress responses, depend on the number of cells with specific responses, rather than on their mean or median, which constitutes the fraction of cells with a given response as a biologically relevant variable. We proposed, therefore, to quantify dose–responses in terms of cellular fractions and show here how this can be achieved for multivariate data.

We first introduced the fractional response curve (FRC) that quantifies fractions of cells that exhibit different responses to a change in dose, or in fact any other experimental condition. For each subsequent dose, the increase of FRC reflects the fraction of cells that exhibit responses different from lower doses. Adding cumulatively distinct fractions results in counting the number of distinct response distributions.

For an illustration of FRC, in addition to the formal definition derived in Methods, we considered a simple hypothetical example involving one signaling effector and three doses, although the approach extends to a general multivariate scenario. Response distributions to three doses, $x_1$, $x_2$, $x_3$, which can be interpreted as control, intermediate, and high dose, are shown in Fig. 2a. When dose 1 was considered alone, fractions of cells with all possible responses sum up to 1 (Fig. 2b). Therefore, we defined the value of the FRC for dose 1 to be 1, and write $r(x_1) = 1$. We then asked what fraction of the cellular population exhibits different responses after the change from dose 1 to dose 2. The fraction of cells exhibiting different responses is equivalent to the overall increase in the frequency of responses (Fig. 2c, green region). The overall fractional increase, denoted as $\Delta r$, is calculated as the area of the green region, and $\Delta r = 0.31$, represents the 31% of the cellular population exhibiting different responses due to dose increase. Therefore, we defined the value of the FRC for dose 2 to be the sum of the previous value and the fractional increment, $r(x_2) = r(x_1) + \Delta r = 1.31$. When dose 3 was considered, the fraction of cells that exhibited different responses is again equivalent to the overall increase in the frequency of different responses, now compared with the two lower doses (Fig. 2d). As before, the overall increase, $\Delta r$, is equivalent to the area of the yellow region (Fig. 2d), with $\Delta r = 0.74$, representing 74% of cells stimulated with dose 3 exhibiting responses different to populations stimulated with lower doses. Again, the value of the FRC for dose 3 was defined as the sum of the previous value and the fractional increment, $r(x_3) = r(x_2) + \Delta r = 2.05$. Changes in the FRC show what fraction of cells exhibit different responses owing to the dose increase. Adding subsequent fractional increments, $\Delta r$, leads to the value of FRC expressed in terms of the cumulative fraction of cells that exhibit different responses due to dose change.

The sum of the dose-to-dose increments, also, records the number of distinct response distributions that were

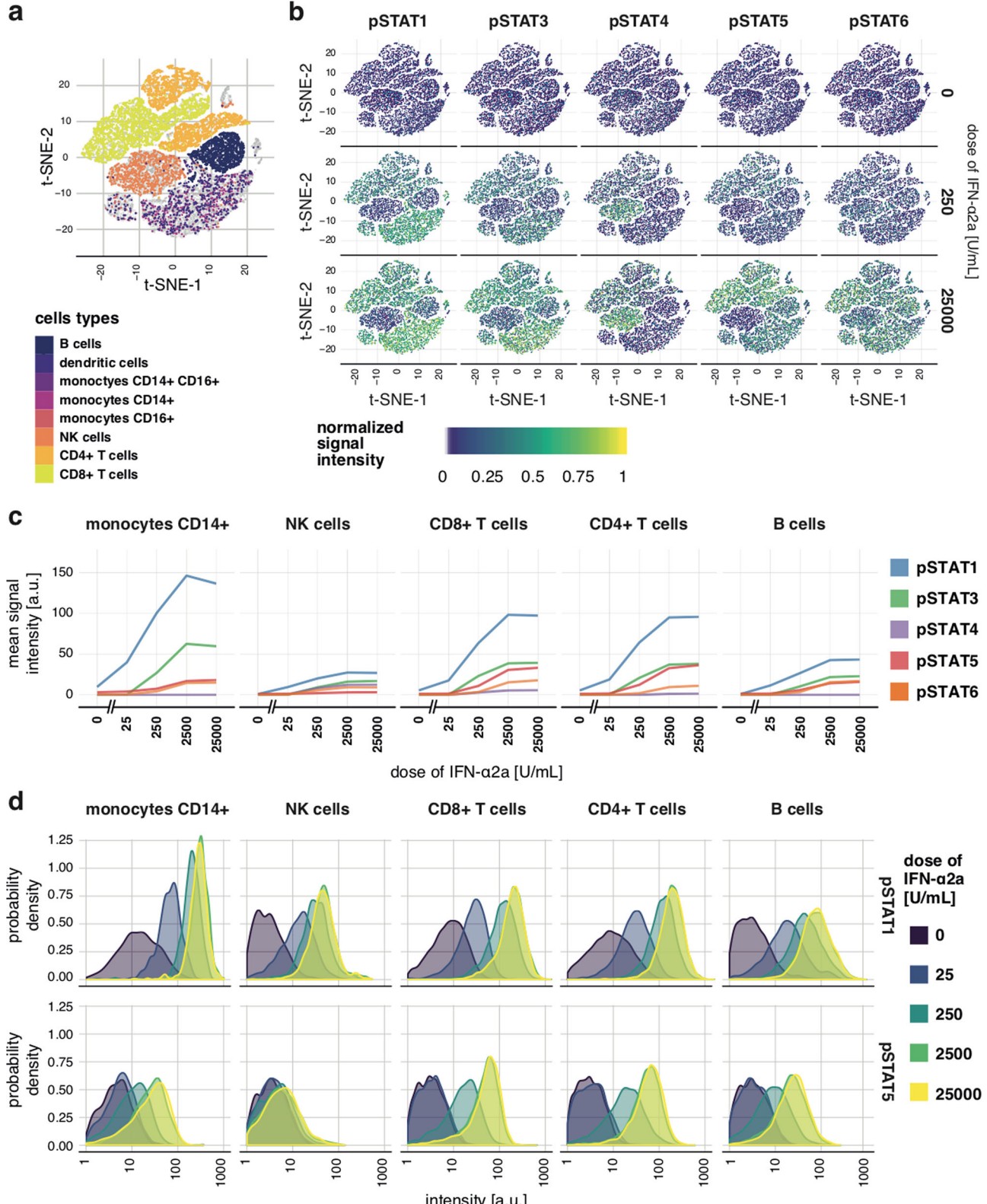

experimentally observed, which provides the second interpretation of the FRC. Precisely, for dose 1 considered alone, a single response distribution was observed, $r(x_1) = 1$. Dose 2 added 31% of a distinct distribution, and $r(x_2) = 1.31$ (the gray area, Fig. 2d). Similarly, accounting for all three doses we had 2.05 distinct response distributions (the gray area, Fig. 2e). The number of distinct response distributions induced by changing dose

quantifies the number of programmed responses of a cellular population, which appears to provide relevantly, yet, so far, unexplored, quantitative characteristics of signaling systems (Supplementary Fig. 5).

The FRC can be universally calculated for any type of signaling data, i.e., an arbitrary number of signaling effectors, time points of measurements, doses, or other experimentally varied

**Fig. 1 Dose–responses to IFN-α2a in PBMCs. a** t-SNE plots constructed based on phenotypic markers. Cell types are encoded by color and each dot represents a single cell. **b** t-SNE plots of whole-cell pSTATs levels 15 min after stimulation with two selected doses of IFN-α2a as well as in unstimulated cells. Positions of dots corresponding to single cells are the same as in **a** allowing cell type identification. The color of each dot represents normalized (0 for minimum and 1 for maximum) mass cytometry signal. Analogous t-SNE plots for all considered doses are shown in Supplementary Fig. 3. **c** Mean pSTATs levels in five cell types as a function of the dose calculated from mass cytometry signals of single cells. The mean of log-data and medians are shown in Supplementary Fig. 4a, b. **d** Distributions of responses in five cell types after stimulation with different doses of IFN-α2a in terms of pSTAT1 (top row) and pSTAT5 (bottom row) as measured with mass cytometry. The shown probability density is proportional to the frequency of cells with a given level of the pSTAT. The value of the probability density is proportional to the frequency of cells with given response levels. Distributions of other pSTATs are shown in Supplementary Fig. 4c. Different doses correspond to different colors. Technical details: at least 2500 cells were measured per condition. The plot shows one representative of two biological replicates.

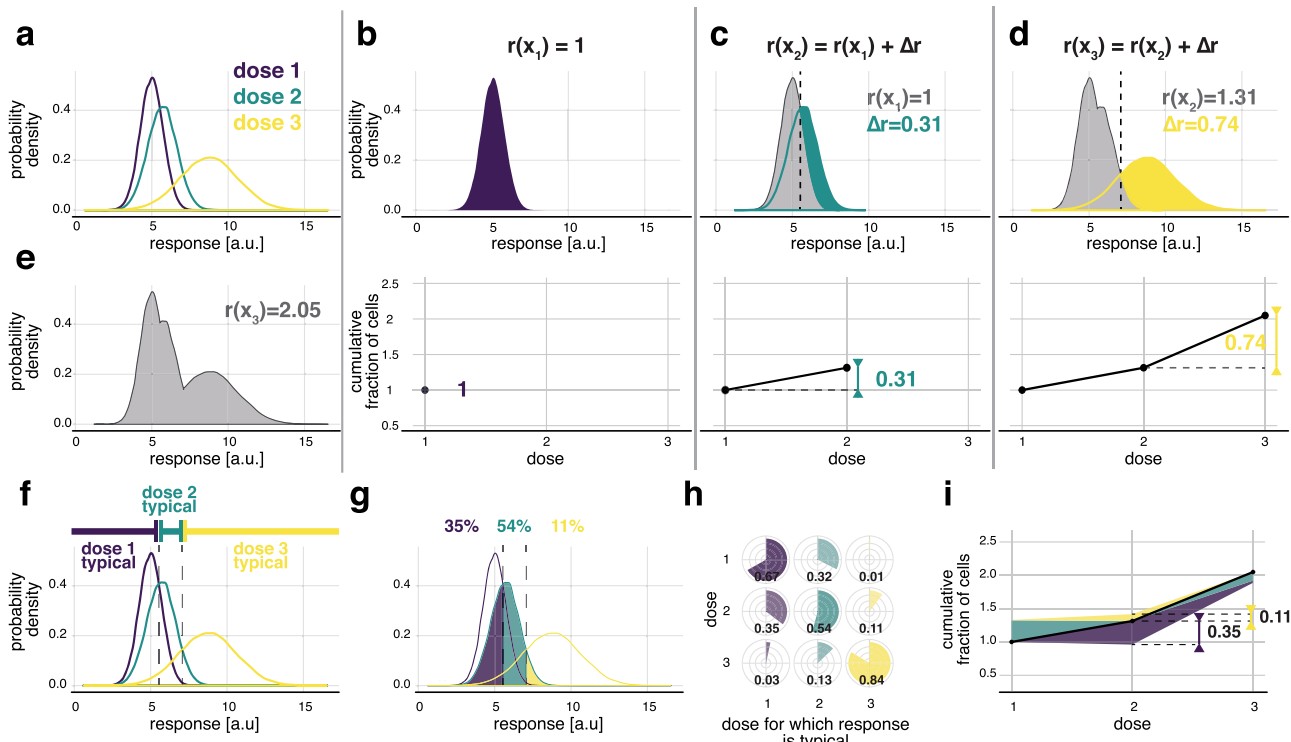

**Fig. 2 Fractional response analysis. a** Hypothetical response distributions to three different doses encoded by colors. Distributions are represented as a probability density, which is proportional to the frequency of cells with a given response level. **b**–**d** Quantification of the fraction of cells that exhibit different responses due to dose increase, Δr, and constriction of FRC, for responses presented in **a**. Each panel from **b** to **d** corresponds to subsequent changes in dose. The color regions mark the overall increase in frequency due to considering the dose marked by the color. The area of the colored region quantifies Δr. The value of the FRC for each dose is obtained by adding the increment, Δr. **e** Quantification of the number of distinct distributions induced by the three considered doses. **f** Dose-typical responses for the response distributions of **a**. **g** Dissection of the responses to dose 2 into responses typical to any of the three doses. The fraction of cells typical to a given dose is marked with the corresponding color. The surface area of each color quantifies the typical fraction. **h** The fractions of cells stimulated with one dose (rows) with responses typical to any of the doses (columns). **i** The FRC together with the bands representing cell-to-cell heterogeneity as quantified in **h**. For each reference dose (x- axis), the fractions of cells stimulated with the reference dose that exhibit responses typical to other doses can be plotted in the form of color bands around the curve. The color encodes the dose a given fraction refers to. The height of the band marks the size of the fraction (y-axis). Fractions corresponding to doses higher than the reference dose are plotted above the curve, whereas to doses lower than the reference dose below the curve.

parameters, as long as sufficient data are available. The interpretation for univariate and multivariate data are the same (compare Fig. 2 and Supplementary Fig. 6). The response probability distributions do not need to be explicitly quantified, as the distinct fractions can be estimated with logistic regression (see Methods) or, in principle, other statistical classifiers, which is particularly relevant for multivariate data.

In addition to the interpretations in terms of fractions of cells, FRC has a rigorous mathematical definition in terms of Rényi information, which, broadly speaking, counts probability distributions corresponding to outputs of a communication system (see Supplementary Note 1 and Supplementary Table 2). It differs from more frequently used Shannon information as discussed in

detail in Supplementary Note 2 using G protein-coupled receptors (GPCRs) signaling data[6] as well as theoretically.

**Fractional cell-to-cell heterogeneity.** The FRC quantifies fractions of cells that exhibit different responses due to dose change but does not quantify overall cell-to-cell heterogeneity: it does not show what fraction of cells exposed to one dose exhibits responses in the range characteristic for other doses. Therefore, within FRA, we propose to augment the FRC with quantification of the overlaps between distributions corresponding to different doses. We call a given response as typical for a given dose if it is most likely, i.e., most frequent, to arise for this specific dose compared

to all other doses. In the hypothetical example, low responses are most likely, and therefore typical, for dose 1, intermediate responses are typical for dose 2, and high responses for dose 3 (Fig. 2f). We can then divide responses to a given dose into responses typical for any dose. For instance, for dose 2, 35% of cells have responses typical for dose 1, 54% typical for dose 2, and 11% typical for dose 3 (Fig. 2g). The results, presented as pie-charts, can be shown in a matrix as the fraction of cells stimulated with one dose (rows) that has responses typical for other doses (columns) (Fig. 2h). This pie-chart partitioning can be plotted along with the FRC (Fig. 2i) so that the fractional increments, $\Delta r$, and fractional cell-to-cell heterogeneity are concisely presented. Similar to FRC, quantification of the fractional cell-to-cell heterogeneity structure can be performed for multivariate data without quantification of the response distributions (see Methods and Supplementary Fig. 6).

**Populations of different types of PBMCs exhibit very similar logarithmic dose–responses.** FRA compresses complex dose–response data into a simple quantitative description accounting for cell-to-cell heterogeneity and multivariate measurements. To determine the kinds of biological information that can be uncovered, we performed FRA for IFN-α2a multivariate dose–responses in different types of PBMCs, assuming that all five measured pSTATs jointly constitute a cell's response. The FRC and fractional cell-to-cell heterogeneity (Fig. 3a, b) are very similar for all cell types. Counter to the differences seen in the analysis presented in Fig. 1a–d and Supplementary Fig. 4, the dose–responses in different cell types follow the same logarithmic pattern identifying a phenomenon that governs the behavior of multivariate cellular responses of our system, which remains hidden when inspecting data in the conventional way.

For all cell types, the FRC is linear and increases at the same rate with respect to the log of the dose, which means that the fraction of cells showing different responses is proportional to the dose fold-change, over a broad range of doses, i.e., from 0 to 2500 U/mL. The linear increase of the FRC demonstrates that the fraction of cells that exhibit different responses are very similar from 0 to 25 U/mL, from 25 to 250 U/mL, and from 250 to 2500 U/mL. For each subsequent dose change, $\Delta r \approx 0.5$ so that 50% of cells have different responses. A given fold-change in the dose induces a different response in the fixed fraction of cells, across a broad range of doses. Therefore, cellular populations are sensitive to fold-changes in the dose as opposed to additive changes.

Formally, FRC scales as the log of the dose

$$r(x) \propto \log(x), \qquad (1)$$

which given incremental approximation, $\Delta\log(x) = \log(x + \Delta x) - \log(x) \approx \Delta x/x$, implies fold-change sensitivity in the population

$$\Delta r \propto \Delta x/x, \qquad (2)$$

which in the studied system universally describes dose–responses in populations across different cell types.

The FRA, therefore, condenses the description of the complex multivariate responses into a simple quantitative formula. Furthermore, FRA uncovered that the number of programmed response distributions, i.e., maximal value of FRC, and the fractional cell-to-cell heterogeneity structure are very similar for all cell types. This similarity indicates that the immune system may precisely control responses of fractions of cells rather than responses of individual cells. In multicellular organisms, a fraction of cells with a given response level is a biologically essential response variable. For example, the outcome of a viral infection in tissue depends on the number of NK cells with given

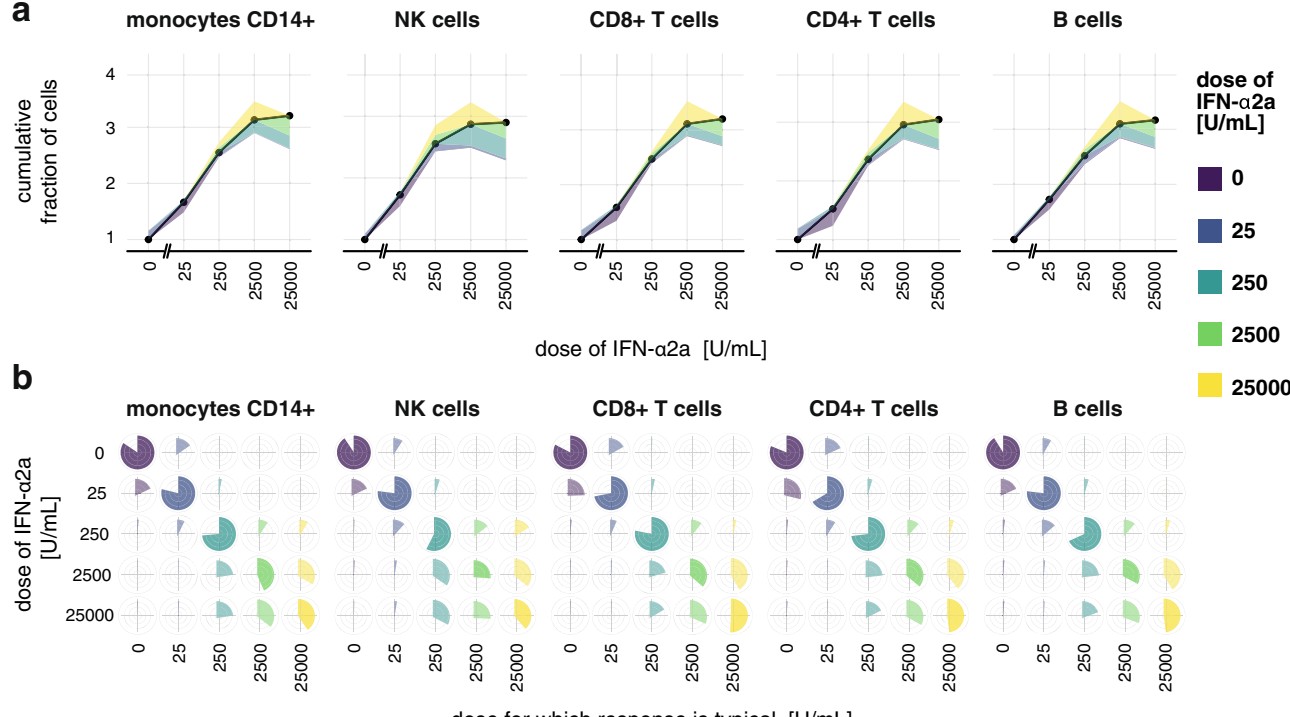

**Fig. 3 Different types of PBMCs exhibit highly similar dose–responses to IFN-α2a. a** FRA of IFN-α2a responses. Here, levels of all pSTATs were assumed to jointly constitute cell's response. Supplementary Fig. 7 shows FRA for individual STATs. **b** Pie-charts of the cell-to-cell heterogeneity structure used to plot color bands in **a**. Cell-to-cell heterogeneity is shown as pie-charts, in addition to **a**, in order to clearly visualize the similarity between the cell types. Technical details: The plot presents an analysis of one biological replicate. The analysis of the second biological replicate yielding very similar results is shown in Supplementary Fig. 8.

response levels and induced cytotoxic activity. Our analysis revealed that in the studied system the fraction of cells that have responses in a specific range is not only tightly controlled in the population of a given cell type but is controlled in the same way across different cell types, as opposed to responses of individual cells that are largely heterogeneous within one cell type and across cell types. The role for controlling the fractions of cells with specific responses can, in principle, be tested by perturbing cell-to-cell heterogeneity through genetic manipulation and observing the phenotypic effects on the performance of the immune system.

**Fold-change sensitivity of cellular populations is a recurrent property of cytokine signaling.** To explore how responses that are qualitatively different translate into differences in FRA, we examined responses to the cytokines interferon-gamma (IFN-γ) and interleukin 10 (IL-10). As these are implicated in macrophage phenotypic diversity[34], we used the human monocyte cell line U937, differentiated into macrophage-like cells, and immunostained to measure responses via nuclear levels of the key signaling effectors, phosphorylated STAT1 for IFN-γ, and phosphorylated STAT3 for IL-10 at 30 min after stimulation (Fig. 4a, b). For IFN-γ, response distributions shift gradually towards higher values as the dose increases, which is referred to as the graded response[35–37]. For IL-10, the distributions flatten over a broad region as the dose increases reflecting the higher number of responding cells for high doses, with the dose having a limited impact on the level of the response, similar to a binary system[2,37] where responses aggregate in "on" and "off" regions. The qualitative differences in the responses to IFN-γ and IL-10 cytokines are mirrored by FRA (Fig. 4d, e and Supplementary Fig. 11a, b). Compared with IL-10, FRC for IFN-γ increases faster and reaches a higher value, which reflects the higher number of cells with distinct responses for increasing doses. Besides, bands around FRC for IFN-γ are narrower than for IL-10, indicating that the response distributions are more distinct. Furthermore, for IL-10 the bands below the curve are broader than above the curve, which reflects the large fractions of cells with response typical to doses lower than encountered, which points to the similarity with the binary system. In Supplementary Note 3, we used in silico generated data of exact binary and graded responses, as well as responses of lung cancer cell lines to IFN-γ to show in detail how FRA can discriminate between different response modalities. Overall, differences in the response distributions visible to the naked eye are adequately mapped onto the FRA plot, which

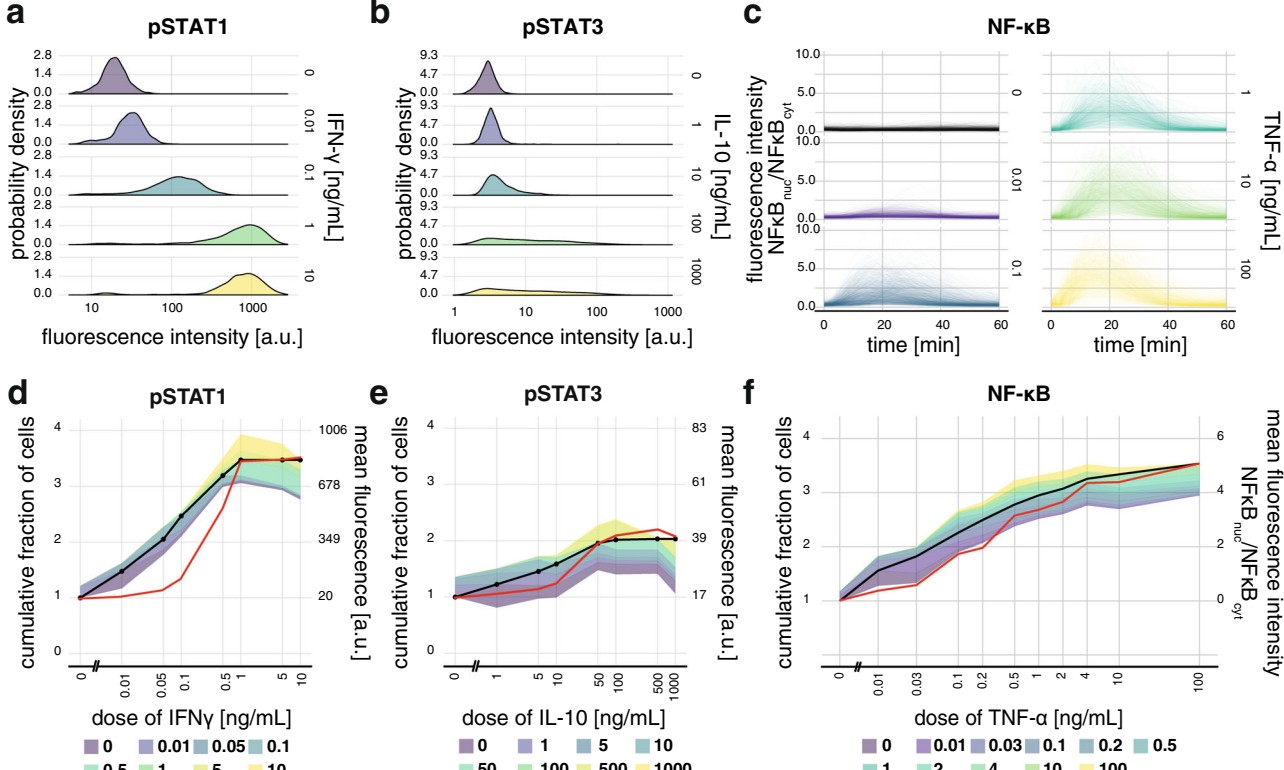

**Fig. 4 IFN-γ, IL-10, and TNF-α exhibit logarithmic dose–responses in cell lines. a** Distributions of responses 30 min after stimulation with different doses of IFN-γ in terms of nuclear pSTAT1 as measured with confocal microscopy imaging and immunostaining in the U937 cell line. Responses are expressed as mean fluorescence of nuclear pixels. Selected doses are shown. See Supplementary Fig. 9 for all doses. **b** The same as in **a** but for pSTAT3 after stimulation with IL-10. **c** Temporally resolved responses of individual murine embryonic fibroblasts to increasing concentrations of TNF-α. Each line corresponds to a single cell. Responses are expressed as the ratio of the nuclear pixels fluorescence average to the cytoplasmic-pixels fluorescence average. Selected doses are shown in the panel. See Supplementary Fig. 10 for all doses. Measurements were taken every 3 min in a murine embryonic fibroblast cell line stably expressing the p65 protein component of the NF-κB complex fused with the fluorescent dsRed protein. **d** FRA of IFN-γ responses. For comparison, the red line presents the mean response with y-axis on the right. The pie-chart corresponding to the plotted cell-to-cell heterogeneity structure is shown in Supplementary Fig. 11. Comparison with mean of log-data is shown in Supplementary Fig. 12. **e** Same as in **d** but for IL-10 responses. **f** FRA of the temporally resolved responses to TNF-α. The red line presents the mean response at the time of maximal response, i.e., at 18 min. Comparison with mean of log-data at 18 min is shown in Supplementary Fig. 12. Technical details: at least 892 cells were measured per condition. Experiments were performed in two biological repeats each consisting of two technical replicates. In **a** and **b**, the sum of two technical replicates of a representative biological replicate is shown. In **c** sum of biological replicates is shown[26].

indicates that similar phenomena hidden by the complexity of multivariate data could be uncovered in the same way.

To further explore how generally applicable FRA is, we examined time-series responses to cytokine stimulation. We measured TNF-α dose–responses with live confocal imaging of a murine embryonic fibroblast cell line stably expressing fluorescent NF-κB complex[4], a key TNF-α signaling effector[4,29], for 60 min (Fig. 4c). As the data constituting the time-series are multivariate, it is not feasible to assess the fraction of cells with distinct responses for increasing doses as well as overlaps between response distributions using visual inspection alone. FRA, on the other hand, enables the quantification and visualization of the cell-to-cell heterogeneity structure (Fig. 4f and Supplementary Fig. 11c). The overlaps between distributions are considerable. The rate of the increase of FRC, as well as width of the bands around FRC, are more similar to IFN-γ than for IL-10, which cannot be seen here directly from the time-series data (Fig. 4c). Primarily, however, FRC increases linearly with the log of TNF-α dose, which again discloses the fold-change sensitivity of the cellular population.

Despite differences in the responses to the above cytokines and type of data used, FRC increases almost linearly with respect to the log of the dose for all three cytokines. Therefore, similarly to IFN-α2a in PBMCs, IFN-γ, IL-10, and TNF-α responses are sensitive to fold-changes in the dose, as opposed to additive changes, suggesting that this mode of the response may be a more universal biological pattern that describes cytokine signaling in cellular populations.

## Discussion

Sensitivity to dose fold-changes in populations of cells resembles the empirical Weber-Fechner law that characterizes the performance of many psycho-physiological sensory systems. Minimal detectable stimulus change, Δx, in the sense of weight, hearing, vision, and smell, has been observed to be of fold-type. So far, several pathways have been shown to follow some form of fold-change sensitivity either by observing representative individual cells or population averages. For instance, Wnt and TGF-β signaling exhibit desensitization to background ligand concentration and subsequent sensitivity to fold-changes[38,39] with an incoherent feed-forward loop motif being the explanatory mechanism[40–42]. Also, single-cell gene expression induced by the nuclear signaling effectors β-catenin, SMAD, or NF-κB, were shown to be sensitive to fold-changes in their nuclear levels[11,38,39]. Similarly, changes in inter-spike intervals in $Ca^{2+}$ spike trains are proportional to baseline inter-spike intervals in GPCRs signaling activated with phospholipase C ligands[43].

Here, FRA allowed us to make a considerably different observation. We demonstrated that within heterogeneous populations of cells of a given type, and across types, the number of cells that exhibit a different response is proportional to the fold-change in the dose. We did not refer to a single signaling effector in a representative cell or population average but to the state of the heterogeneous population described by multivariate data. Ultimate outcomes of multicellular processes like immunity are not determined by individual cells alone or population averages but by a heterogeneous collective. By accounting for cell-to-cell heterogeneity, we showed that the distribution of the collective, which encodes stimulation levels in multicellular systems, shifts with the fold-changes of the dose. Therefore, the way in which heterogeneous cell populations encode signals is quantitatively similar to the way we perceive differences in certain sensations (weight/light).

Weber-Fechner law is a pattern that can arise from a range of different mechanisms[44,45] with the underlying neural

implementations still being discovered[45,46]. Here also, a mechanistic explanation of the fold-change sensitivity of cellular populations is not clear and remains to be determined, possibly by relating cell-to-cell heterogeneity with ligand sensitivity, which might involve feed-forward loops, as in fold-change detection in individual cells over a long time scale[40–42].

Overall, FRA delivers a concise representation of complex single-cell data, which is particularly relevant for high-throughput techniques, which are increasingly allowing the measurement of a high number of parameters per cell, generating very large, high-dimensional datasets[47]. The high information content of multivariate, single-cell measurements makes biological discoveries more likely. On the other hand, however, insights may be difficult to extract due to data complexity. Therefore, making use of the increasing amount of single-cell high-throughput measurements requires approaches that can extract relevant insights in spite of complexity. FRA is not limited to cytokine signaling, proteomic data, or dose–responses, enabling the systematic investigation of single-cell high-throughput data in a wide range of situations, in which responses are measured in single cells at any "-omics" scale. Accurate estimation requires, however, the number of measured cells to considerably exceed the number of measures signaling effectors, and a representative selection of doses (see Supplementary Note 4 for caveats of FRA). Nonetheless, FRA should yield insights into the structure of signaling heterogeneity in immunology, developmental biology, cancer research, and diverse other fields in which response analysis in single cells is of relevance.

## Methods

**Software implementation**. The methodology to perform and visualize FRA is provided as a user-friendly R-package available for download at http://github.com/sysbiosig/FRA. The package contains an installation guide and a brief user manual.

**Formal definition of the FRC**. Consider a series of doses $x_1,...,x_i,...,x_m$ and denote a single-cell response as $y$. Depending on the context, $y$, may be a number or a vector, e.g., the level of one or more measured signaling effectors. Suppose that responses to a given dose, $x_i$, are represented as the probability distribution,

$$P(Y|x_i). \tag{3}$$

The FRC is then formally defined as

$$r(x_i) = \int_{\mathcal{Y}} \max_{x_k \le x_i} P(y|x_k)dy, \tag{4}$$

where integration takes place over $\mathcal{Y}$, the set of all possible responses, $y$. The integral quantifies the area under the curve (or under surface for multivariate data), with respect to $y$, defined as $\max_{x_k \le x_i} P(y|x_k)$. For the calculations shown in Fig. 2 the integration corresponds to the calculation of the area of the gray regions in c–e. As explained in Supplementary Note 1, the FRC defined as above is closely related Rényi min-information capacity.

**Formal definition of typical fractions**. Having the responses represented in terms of the probability distribution, Eq. 3, we can define which responses, $y$, are typical to any of the doses. Precisely, we define the response, $y$, to be typical for dose $x_j$ if it is most likely to arise for this dose, which writes as

$$P(y|x_j) > P(y|x_k) \text{ for all } k \text{ other than } j. \tag{5}$$

The above condition allows assigning any response, $y$, to a dose for which it is typical. Therefore, for a given dose, $x_i$, we can identify what fraction of cells stimulated with this dose exhibits responses typical to any dose, $x_j$, for $j$ from 1 to $m$. These fractions, denoted as $v_{ij}$, can be practically computed as explained below.

**Calculation of typical fractions**. The fractions of cells stimulated with dose $i$ that have responses typical to dose $j$, $v_{ij}$, can be easily calculated from data regardless of the number of doses and the type of experimental measurements. We have that

$$v_{ij} = \frac{\text{number of cells stimulated with } x_i \text{ with responses typical for } x_j}{\text{number of cells stimulated with } x_i}. \tag{6}$$

Calculation of typical fractions, $v_{ij}$, with the above formula requires the possibility to examine the condition $P(y|x_j) > P(y|x_k)$ for any experimentally observed response, $y$. The distributions $P(y|x_j)$ can be reconstructed from data using a variety of probability density estimators[48]. The use of the available estimators,

however, might be problematic for multivariate responses[26,49]. We, therefore, propose a more convenient strategy. We replace the condition $P(y|x_j) > P(y|x_k)$ with an equivalent condition that is computationally much simpler to evaluate. Precisely, we propose to use the Bayes formula

$$P(x_j|y) = \frac{P(y|x_j)P(x_j)}{\sum\limits_{k=1}^{m} P(y|x_k)P(x_k)}. \tag{7}$$

If we set the equiprobable prior distribution, i.e., $P(x_j) = 1/m$, we have that $P(y|x_j)$ is proportional to $P(x_j|y)$ and the condition $P(y|x_j) > P(y|x_k)$ is equivalent to

$$P(x_j|y) > P(x_k|y). \tag{8}$$

The above strategy allows avoiding estimation of the response distributions, $P(y|x_j)$, from data. For continuous and multivariate variable $y$ the estimation of $P(x_j|y)$ is generally simpler than estimation of $P(y|x_j)$[26,48]. Precisely, an estimator $\hat{P}(x_j|Y=y)$ of the distribution $P(x_j|y)$ can be built using a variety of Bayesian statistical learning methods. For simplicity and efficiency, here we propose to use logistic regression, which is known to work well in a range of applications[48]. In principle, however, other classifiers could also be considered. The logistic regression estimators of $P(x_j|Y=y)$ arise from a simplifying assumption that log-ratio of probabilities, $P(x_j|Y=y)$ and $P(x_m|Y=y)$ is linear. Precisely,

$$\log\left(\frac{P(x_j|Y=y)}{P(x_m|Y=y)}\right) \approx \alpha_j + \beta_j^T y. \tag{9}$$

The above formulation allows fitting the logistic regression equations to experimental data, i.e., finding values of the parameters, $\alpha_j$ and $\beta_j$ that best represent the data. The fitted logistic regression model allows assigning cellular responses to typical doses based on conditions given by Eq. 8. Formally, the fractions $v_{ij}$ defined by Eq. 6 are calculated as

$$v_{ij} = \frac{1}{n_i} \sum\limits_{l=1}^{n_i} \mathbb{1}_{\left\{\hat{P}(x_j|Y=y) > \hat{P}(x_k|Y=y) : k \neq j\right\}}(y_i^l), \tag{10}$$

where $n_i$ is the number of cells measured for the dose $x_i$, $y_i^l$ denotes response of the $l$-th cell, and $\mathbb{1}_{\left\{\hat{P}(x_j|Y=y) > \hat{P}(x_k|Y=y) : k \neq j\right\}}(y_i^l)$ is equal 1 if $\hat{P}(x_j|Y=y) > \hat{P}(x_k|Y=y)$ for any $k \neq j$ and 0 otherwise.

**Calculation of the FRC**. Calculation of the FRC can be conveniently performed using the typical fractions, as defined above, rather than through integration of Eq. 4. Precisely, to calculate the FRC for the dose, $x_i$, consider doses $x_1,...,x_i$ in isolation from higher doses. Then, the sum of typical fractions $v_{11},...,v_{ii}$ is equivalent to FRC for the dose $x_i$

$$r(x_i) = \sum\limits_{k=1}^{i} v_{kk}. \tag{11}$$

The equivalency of the above equation and Eq. 4 is derived in the Supplementary Methods.

**Mass cytometry (CyTOF)**. PBMCs were isolated from the peripheral blood of healthy adult donors using Lymphoprep (Stemcell Technologies), according to the manufacturer's instructions. Cells were washed in serum-free Roswell Park Memorial Institute (RPMI) then resuspended at $10^7$ cells/mL in serum-free RPMI containing 0.5 mM Cell-ID Cisplatin (Fluidigm) and incubated at 37 °C for 5 min. Cells were washed with RPMI containing 10% (v/v) FCS (Sigma) and 2 mM L-Glutamine (R10), centrifuging at $300 \times g$ for 5 min before being resuspended to $6 \times 10^7$ cells/mL in R10 and rested at 37 °C for 15 min. 50 mL of cells ($3 \times 10^6$ cells) were transferred to 15 mL falcon tubes for stimulation and antibody staining. Antibodies and their dilutions are listed in Supplementary Table 1. Staining for CD14, CCR6, CD56, CD45RO, CD27, CCR7, CCR4, and CXCR3 was done before stimulation/fixation for 30 min in R10 at 37 °C. Cells were stimulated with 0, 25, 250, 2500, or 25000 U/mL recombinant human IFN-α2a (PBL Assay Science, #11100-1) diluted in R10 for 15 min at 37 °C. After washing with 5 mL cold Maxpar PBS (Fluidigm), cells were fixed with 1× Maxpar Fix I Buffer (Fluidigm) for 10 min at RT before being washed with 1.5 mL Maxpar Cell Staining Buffer (CSB, Fluidigm). All centrifugation steps after this point were at $800 \times g$ for 5 min. Cells were barcoded using Cell-ID 20-Plex Pd Barcoding Kit (Fluidigm), according to the manufacturer's instructions, and washed twice with CSB before samples were pooled and counted. All further steps were performed on the pooled cells. Fc receptors were blocked using Fc Receptor Binding Inhibitor Antibody (eBioscience, #14-9161-73) diluted 1:10 in CSB for 10 min at RT. Surface antibody staining mixture was added directly to the blocking solution and incubated for 30 min at RT. Cells were washed twice with CSB, resuspended in ice-cold methanol, and stored at −80 °C overnight. After washing twice with CSB, cells were stained with intracellular antibody staining mixture for 30 min at RT before two further washes in CSB. Cells were resuspended in 1.6% (v/v) formaldehyde (Pierce, #28906) diluted in Maxpar PBS and incubated for 10 min at RT. Cells were resuspended in 125 mM Cell-ID Intercalator (Fluidigm) diluted in Maxpar Fix and Perm Buffer (Fluidigm) and incubated overnight at 4 °C. Compensation beads (OneComp eBeads Compensation Beads, Invitrogen, #01-1111-42) stained with 1 mL of each antibody were also

prepared. The next day, cells and compensation beads were washed twice with CSB and twice with Maxpar water (Fluidigm), mixed with a 1:10 volume EQ Four Element Calibration Beads (Fluidigm) before acquisition on a Helios Mass Cytometer (Fluidigm) using the HT injector. Data were normalized, randomized, and concatenated using Helios CyTOF Software v6.7 (Fluidigm), cytofCore v0.4, flowCore v1.46.2, and Cytobank v6.2. Compensation and de-barcoding were performed using the CATALYST v1.5.3.23 package[50]. Different immune cell subpopulations were gated in R v3.5.1 from single, live, CD45+ cells as shown in Supplementary Fig. 2.

Collection and analysis of PBMCs were carried out in accordance with the EU Directive 2004/23/EC and the UK Human Tissue Act 2004 (HTA), under the HTA licence (number 12433) of the Weatherall Institute of Molecular Medicine. Informed consent was obtained and the samples were fully anonymised.

**U937 cells**. U937 cells (CRL-1593.2, ATCC), a human monocyte cell line, were cultured under standard conditions at 37 °C in a humidified atmosphere of 5% $CO_2$/95% air in low glucose RPMI 1640 (Corning, #10-040-CV) medium supplemented with 10% fetal bovine serum (FBS, ThermoFisher, #10500064) and 1% penicillin–streptomycin solution (P/S, ThermoFisher, #15140122). For macrophage differentiation, U937 cells were suspended in a medium with 20 ng/mL phorbol 12-myristate 13-acetate (PMA, Sigma Aldrich, #P1585) and plated in 96-well microplates with μClear® flat bottom (Greiner, #655090) in density $2 \times 10^4$ cells per well. After 24 h medium with PMA was removed and fresh medium was added to cells. 72 h after seeding on 96-well microplates differentiated cells were incubated with recombinant human IFN-γ (ThermoFisher, #PHC4031) at concentrations 0DAPI10 ng/mL or recombinant human IL-10 (PeproTech, #200-10) at concentrations 0–1000 ng/mL for 30 min. Afterwards, cells were fixed with 3.7% paraformaldehyde (PFA, Sigma Aldrich, #P6148) for 10 min at room temperature, RT, then permeabilized with 90% ice-cold methanol (Sigma, #322415), for 30 min at −20 °C, blocked with 5% bovine serum albumin (BSA, Merck, #821006) and 0.3% Triton X-100 (Sigma Aldrich, #T9284) for 1 h at RT, and incubated with primary antibody–phospho-STAT1 (Tyr701) (pSTAT1, Cell Signaling, #9167) diluted 1:100 or phospho-STAT3 (Tyr705) (pSTAT3, Cell Signaling, #4113) diluted 1:200 in 1% BSA with 0.3% Triton X-100 for 18 h at 4 °C. Next day, cells were incubated with an appropriate secondary antibody–Alexa Fluor 488 (Life Technologies, #A-21206) or Alexa Fluor 555 (Life Technologies, #A-31570) diluted 1:500 in 1% BSA with 0.3% Triton X-100 for 1.5 h at RT and stained with 2 μg/mL 4',6-diamidino-2-phenylindole (Sigma Aldrich, #D9542) for 10 min at RT. The fluorescence signal was acquired using an automated confocal microscope (Pathway 435, BD) and analyzed with Cell Profiler v2.1.1, R v3.5.1, and ImageJ v1.48.

**Murine immortalized fibroblasts**. Murine embryonic fibroblasts 3T3 cell line, previously used in several studies including[4,26], expressing fluorescent fusion proteins relA-dsRed as wells H2B-GFP for nuclei identification were cultured in an incubator under standard conditions at 37 °C in a humidified atmosphere of 5% CO2/95% air. The cell line was kindly provided by Professor S. Tay. The cells were cultured in high glucose Dulbecco's Modified Eagle's Medium without phenol red (ThermoFisher, #21063029) supplemented with 10% FBS (ThermoFisher, #10500064) and 1% penicillin–streptomycin solution (P/S, ThermoFisher, 15140122). Approximately $1.3 \times 10^5$ cells were plated on 35-mm confocal dish for imaging. After 48 h in the incubator, cells were transferred to the environmental chamber in a microscope. At time 0, medium was removed from cells and recombinant mouse TNF-α (Sigma Aldrich, #T7539) was added at concentrations 0-100 ng/mL as a 5-minute pulse. Live imaging was performed using a confocal microscope, Leica TCS SP5 X. During single experiment, images have been captured every 3 min over 1 h in two channels simultaneously at nine different positions on the plate. The experiment has been repeated at least four times to test reproducibility and to allow for a sufficient number of observations. Nuclear and cytoplasmic fluorescence (pixel mean) was then quantified from microscopic images using Cell Profiler v2.1.1, and R v3.5.1. The response of each cell was then represented as the ratio of nuclear to cytoplasmic fluorescence in order to ensure the robustness of measurements to changes in confocal plane over time. The data set is described in detail in ref. [26], where it was initially published.

**Reporting summary**. Further information on research design is available in the Nature Research Reporting Summary linked to this article.

## Data availability
Data generated during the study are available for download from https://github.com/sysbiosig/FRA/ and are also deposited in the open-access repository https://doi.org/10.5281/zenodo.4835622. The study also involves data published in ref. [26].

## Code availability
FRA is made available as an R-package (see Supplementary Note 5) downloadable from https://github.com/sysbiosig/FRA/ that is also deposited in the open-access repository https://doi.org/10.5281/zenodo.4818586.

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

## Acknowledgements

K.N., J.W., K.E.Z., and M.K. were supported by Foundation for Polish Science within the First TEAM (First TEAM/2017-3/21) program co-financed by the European Union under the European Regional Development Fund. K.N. was supported by Polish National Science Centre under grant PRELUDIUM 2016/23/N/ST6/03505. R.E.R. and J.R. were funded by the UK Medical Research Council [MRC core funding of the MRC Human Immunology Unit]. We thank Guy Riddihough for helpful comments during the preparation of this manuscript. M.K. and J.R. also thank Matteo Iannacone, the organizer of an EMBO workshop on Immunology, Bergamo 2017, which had initiated our collaboration.

## Author contributions

Conceptualization of FRA.: K.N., M.K.; interpretation of FRA.: K.N., J.W., M.K.; CyTOF study design and experimentation: R.E.R., J.R.; microscopy study design and experimentation: K.E.Z., E.G.; data interpretation and analysis: K.N., R.E.R., J.W., K.E.Z., J.R., M.K.; implementation of the R-software package and of the user manual: K.N.; figure preparation: K.N., M.K.; writing the manuscript: K.N., K.E.Z., R.E.R., M.K.; editing the manuscript: K.N., R.E.R., J.W., K.E.Z., J.R., M.K.

## Competing interests

The authors declare no competing interests.
