## [Peer Review File · Nature Communications]

REVIEWER COMMENTS

Reviewer #1, an expert in mathematical modelling and systems biology of inflammation (Remarks to the Author):

The authors present a “fractional response analysis” (FRA) framework to quantify responses to signals using a variant of Renyi min entropy, a quantity from information theory and an alternative to the more frequently used Shannon entropy. It is suggested that this formalism has advantages to conventional approaches especially when analyzing high-dimensional single-cell data. As biological applications, the authors performed and analyzed a CyTof experiment with PBMCs after stimulation with IFN-alpha, and a microscopic time-series experiment of U937 macrophage-like cells after stimulation with IFN-g or IL-10. Finally, a published kinetic data set on TNF-stimulated fibroblasts was re-examined. As a mayor conclusion, the authors highlight applicability of the Weber-Fechner law to cellular responses to cytokines.

The FRA framework itself and the application of Renyi entropy to signal transduction in biology is a very interesting and timely idea. Indeed, quantitative methods for analysis of single-cell data are highly needed in the field. Moreover, the authors made an effort to provide fresh single-cell data from highly relevant biological model systems (cytokine-driven pSTAT activation in different cell lines), suitable for application of their new FRA approach. However, the downside is that this approach, using a new method with new data, also limits comparison to existing data sets and analysis methods. Therefore, I would recommend adding analysis of published data sets by FRA and/or providing a fair comparison of FRA to a set of other analysis methods on the new data set. In the presented workflow, this kind of analysis is limited to t-SNE plots and mean intensity plots shown in Fig. 1, which cannot really be compared to FRA (see below for details). In summary, before publication, the authors should definitely clarify and substantiate their claim in the abstract, that FRA “uncover otherwise hidden patterns in single-cell data”.

Major comments:

- The proposed mathematical framework based on Renyi entropy is the most interesting part of the study. However, a number of questions remain open, conceptually as well as regarding interpretation and (lack of) comparison to other measures of channel capacity. What is the advantage of using Renyi min entropy with respect to the more classic Shannon entropy? How does the proposed framework relate to other multiple-input information theory methods, such as those recently published by the same group (Jetka et al., PLoS Comp Biol 2019; Jetka et al., Nat Comm 2018)? Is it possible to give a more intuitive explanation of what FRA actually quantifies – in addition to the more algorithmic description given in Fig. 2?

- Technically, I am wondering whether the proposed FRC quantity fulfills some type of “associative law”. Meaning, if we have measurements at dose x_1 , x_2 , x_3 , I understand that $r(x_2) = r(x_1) + dr_{21}$, and $r(x_3) = r(x_2) + dr_{32}$ (Fig. 2C,D) where dr_{ij} is the area under the curve difference between the response distributions $r(x_i)$ and $r(x_j)$. Now, does it hold true that $r(x_3) = r(x_1) + dr_{31}$? If not, would that not have severe implications for interpretation, and even more so for comparison of different data sets? In more general terms, it appears to me that FRA quantifies a specific experiment rather than a biological system – everything depends on the number and location of considered doses (or other measured signals). That does not mean FRA cannot reveal important insights, but such matters should be clearly discussed in the paper, and compared to other methods.

- The shown “conventional analysis” in Fig. 1, and its relation to the proposed new framework, is enigmatic. P.3, “A single t-SNE plot represents responses in terms of one signaling effector only” – that is not necessarily true, and usually not the purpose of t-SNE plots: as shown in Fig. 1A, the main purpose of t-SNE visualization and related clustering methods is classification of single cells and not dose-response analysis. How could such an analysis be replaced by the proposed FRA method? From what is shown in Fig. 3, it rather seems that a cell type analysis method such as t-SNE is a prerequisite for FRA. Further down: “Typically, to obtain quantitative characteristics ... mean responses ... are plotted” – that is not true, in many standard flow cytometry analysis pipelines, the geometric mean index is used for quantification rather than mean intensity. It would be very interesting to see whether that quantity, and/ or a simple log-transformed intensity value, behaves similar to the FRA in a direct comparison to Fig. 3A.

- In light of the research questions and claims raised in the introduction, the conclusions and the

shown analyses of examples are rather underwhelming. In particular, it is suggested that “conventional dose-response curves to not capture the inherent complexity of single-cell high-throughput data” (p.2), with reference to phenomena such as cell-to-cell heterogeneity or signals that change over time. In contrast to that task list, in the results part, the number of parallel responses analyzed does not exceed 5 (pSTAT1,3,4,5,6), and the paper is largely concerned with analyzing single time points (Fig. 1-3 and Fig. 4A,B,D,E) - only Fig. 4 C,F is concerned with time-series analysis, with very limited conclusions. Again, would it be possible to add applications of FRA to more complex, published data sets, and compare the conclusions to those from the original work?

- The main biological discovery, applicability of Weber’s law to cytokine signaling, is interesting but not that new or surprising as suggested by the authors. The discovery of fold-change detection in eukaryotic signaling goes back (at least) to a series of papers on Wnt signaling published in 2009 in Mol Cell (Goentoro and Kirschner; Goentoro ... Alon; James Ferrell). In contrast to the claim of the authors that “a mechanistic explanation ... remains to be determined”, those papers already proposed an incoherent feed-forward loop motif as a plausible mechanism. More recent work described fold-change detection or Weber’s law in context of NFkB (Lee ... Gaudet, Mol Cell 2014) and Ca(2+) signaling (Thurley et al., Sci Signal 2014), to name only a few. Further, it remains unclear how much the described new information-theoretic framework contributed to that major finding of fold-change detection. Would the methods used in the papers mentioned above, which also analyze single-cell data, fail to discover fold-change detection in the data sets analyzed here?

- P. 6, “for IFN-g, distributions of responses shifted gradually ... For IL-10 the distributions flattened ... similar to a binary system”. From my understanding, this is one of the most interesting observations in the manuscript. Could FRA be used in more general terms to distinguish a gradual from an all-or-none-type of response, or even better to quantify the level of gradual response that can be achieved by a certain signaling pathway? If that is the case, it would be very helpful to compare FRA to “conventional methods” in this regard, such as fitting of the dose-response (possibly in terms of geometric mean index) to a Hill curve.

Minor comments:

- It would be advisable to avoid wordings such as “typically” or “conventional methods” without any reference.
- The t-SNE analysis and cell clustering performed in Fig. 1, and used for further analysis in Fig. 3, should be described in much more detail in the methods section, preferably accompanied by giving access to the source-code for the analysis.
- I did not yet evaluate the provided source code for calculating FRA, which should be done at a later stage of the review process if applicable.

Reviewer #2, an expert in single cell signalling and noise in signalling networks (Remarks to the Author):

In “Fractional response analysis reveals logarithmic cytokine responses in cellular populations,” the authors present a novel tool, namely the fractional response analysis (FRA), to quantify and interpret changes in the heterogeneity of single-cell signaling responses after cell stimulation, and for responses obtained by high-throughput techniques. The scientific question is timely and relevant, as the quantification and interpretation of single-cell responses is a fundamental concern of many investigations in cellular biology. The paper is also interesting in that the authors present high-quality experimental data for multiple different combinations of cell lines and conditions, and they analyze these data using their new metric of fraction response.

The proposed method may potentially reveal new insights into single-cell responses, but as the paper is currently written, the actual benefits and limitations of the fractional response analysis are not adequately described or tested. Without a more rigorous investigation of the metric and its ability to discriminate between different models or different biological circumstances, the importance of this new tool is not clear. More detailed comments are provided as follows:

1) The authors introduce in the main text three main difficulties to interpret single-cell signaling data: 1) cell-to-cell heterogeneity, 2) complex responses given the signaling network, and 3) transient responses. The paper falls somewhat short of addressing these issues in different ways.

1a. Regarding heterogeneity: the authors discuss the limitation of conventional dose-response curves commonly used to report single-cell response analysis. The advantage of the use of FRA vs dose-response curves are clear, but this in itself is not an original observation. One of our main concerns is that the authors don't adequately compare their method existing work that has also sought to integrate stochastic models and single-cell data to improve the quantitative characterization of single-cell responses. A very large number of groups have used noise as an additional feature to discriminate between different cell populations with similar means (see for example the work from: Nicolas Buchler, Hana El Samad, Ramon Grima, Srividya Iyer-Biswas, Mustafa Khammash, Heinz Koepl, Thomas Lipniaki, Andrew Mugler, Brian Munsky, Rob Phillips, Arjun Raj, Jakub Reuss, Abhyudai Singh, Eduardo Sontag, and Alexander van Oudenaarden). While it is understandable that no paper could possibly cite all relevant work on this well-studied topic (and we do recognize that other relevant groups are cited but not discussed in any detail), we would expect to see at least some comparison of the FRA to other metrics to quantify differences in cellular variability from one population to another.

1b. Regarding complexity: the authors state that the proposed FRA is amenable to high-content data (e.g., data in which many different transcription factors or reporters are measured simultaneously), and this claim is the main focus of the final paragraph of the discussion. However, the paper itself focuses only on one-dimensional data (i.e., one distribution at a time). The FRC is very easy to understand in 1D when dealing with one feature at a time, but it is not clear how well the method scales with larger dimensions of data. Would the FRC be defined separately and summed over each marginal (as seems to be the case in the current manuscript), or over the full N-dimensional joint distribution. If the former, then how would one detect or account for dependencies in the data in which one feature is strongly correlated with another? If the latter, then how would the method behave as the number of bins grows exponentially with the number of species and the sampling of these bins becomes extremely sparse?

1c. It was stated that FRA could be performed for dynamic measurements. However, the examples seem only show FRA performed on fixed-time population snapshots per dose. Do the authors believe that FRA could reveal patterns in the temporal evolution of the response as well? If so, how would that be done?

2) It seems that the FRA would be sensitive to the choice of binning strategies. We would expect that if bins are too coarse, then the differences between distributions would become negligible and the FRA would not work. If the bins are too fine, then finite sample sizes will make any two probability densities look unique from one another. This issue is particularly important when dealing with multiple features. For example suppose that the measurements include 10 features, each of which can take on values in 5 separate bins. This would suggest up to 5^{10} different bins – how reproducible would the FRC results be under finite amounts of data? How sensitive are the FRA results to specific choices of bins?

3) The FRA offers a simple method to interpret single-cell signaling responses, but at the same time, it is not very clear what is the resolution power of this methodology.

3a. For example, for pStat1 in Fig. 3-a, all the obtained results look very similar for the different cell types, even when the distributions look quite different among the cell types (Fig. 1D). For pStat5 in Fig. S5, the FRA does reveal a difference between NK cells and the rest (this difference is also obvious in Fig 1D).

3b. The paper seems to focus on discussing the similarity of the FRA for different cells and different conditions, but it is not clear what is the utility of a metric that gives the same result for most situations. Rather, it would be better to focus on when the FRA reveals differences that might have been missed using other methods.

4) Overall, the paper as written does not provide a convincing argument for if/when the FRA should be expected to provide discriminatory insight between different cells or different response types.

4a. The paper would be substantially improved with a more rigorous investigation of the FRA metric in a controlled theoretical or computational framework. Performing the analysis on model-generated data (i.e., with a known ground truth) would enable the authors to better probe what kinds of features can the FRA reveal, and what are the limitations of the analysis. For example, the authors could employ several toy models of stochastic gene regulation (maybe even just using simple bursting gene expression models with different regulatory mechanisms or in different parameter regimes), and demonstrate clearly when the FRA is able to discriminate between different models and when it is not.

4b. With simulated data, the authors could probe the effectiveness of the FRA to discriminate different phenotypes and see how that depends on (i) different types of binning, (ii) different transformations of data (e.g., log or linear space), (iii) different sample sizes, (iv) different amounts or types (e.g., additive, multiplicative) of experimental measurement noise, (v) different ranges or resolutions of stimuli, (vi) different shapes of distributions (unimodal, bimodal), (vii) different numbers and dependencies of feature measurements, etc. While not all of these would need to be addressed, the authors should at a minimum show some examples of effects that are well treated using FRA and ideally they could also discuss some that are limitations of FRA.

5) In Figures 3A, 4 D-F, and S5, the color bands are confusing and do not provide much insight as described. Are these just showing the same fraction information as shown in Fig. 3B? If so, the pie charts are much easier to interpret. Either way, the caption needs to include a better description of the figure.

6) On page 6, the sentence, "On the other hand, TNF- α responses are given as time-series, therefore, their characteristics cannot be directly observed." was not clear. Why does time series data prevent observation of the data characteristics?

We thank reviewers for their kind and inspiring comments, which, we believe, have led to a tangibly improved version of the manuscript. Our attention was primarily drawn to the need for a more comprehensive comparison of our approach with existing tools using a previously published data set. Further, we intended to demonstrate that FRA does not only enable finding unifying patterns in complex data sets but can also reveal differences in cellular responses that cannot be quantified with other available methods. Besides, reviewers' feedback made us aware of the necessity to provide a more accessible interpretation of FRA without an algorithmic reference. The above three issues were raised by both reviewers. Therefore, we address these first in the general response preceding the point-by-point response.

Reviewer comments are in black with underlined essentials, our responses are in blue.

1. Comparison to existing analysis methods using published data sets or synthetic data

In the original manuscript, we compared FRC with mean response curves only. Inspired by the reviewers' comments, we concluded that comparison with Shannon information, medians, and mean responses of log-data (geometric mean) would be the most valuable amendment. For comparisons, we used the data described in the original manuscript. Besides, we added an analysis of a data set on GPCR signaling published in: Keshelava, Amiran, et al. "High capacity in G protein-coupled receptor signaling." *Nature communications* 9.1 (2018): 1-8.

Comparison with Shannon information

In Section 2 of Supplementary Notes, we presented a theoretical comparison of Shannon information and Renyi min-information. Renyi min-information has a natural interpretation in the context of cellular population response data, as opposed to Shannon information that requires long sequences of discrete symbols for rigorous interpretation (Shannon coding theorem). Besides, quantitatively, Renyi min-information is greater than Shannon information. Comparison for CyTOF data is shown in Supplementary Fig 11a, and for IFN- γ , IL-10 and TNF- α responses in Supplementary Fig 11b. The differences in estimated values are moderate. The analysis of the newly added data set from Keshelava et al. (2018) on GPCR signaling, Supplementary Fig. 12, makes advantages offered by the Renyi min-information particularly apparent. Variability of responses of individual cells is substantially smaller than cell-to-cell variability. Therefore, the interpretation of Shannon information in terms of the number of resolvable input concentrations is questionable. On the other hand, the interpretation of Renyi min-information is adequate, as discussed in Supplementary Notes' Section 2.4.

Comparison with medians and means of log responses

In the previous version of the manuscript, FRA for CyTOF data was compared to mean responses in Fig. 1 and Fig. 3. To provide a comparison with medians and means of the log data, these were added to Supplementary Fig. 3, which, however, can only be plotted for individual signaling effectors alone. Therefore, it is difficult to draw any conclusions from comparison with FRA, which accounts for all signaling effectors jointly. Further, unlike FRA, mean/median responses do not uncover any unifying pattern regarding behavior of cellular populations of different cell types.

2. Discriminatory advantages of FRA

In order to demonstrate how FRA can uncover patterns masked by population averages and, hence, discriminate between different response modalities, we considered two examples. An *in silico* model was tailored to generate graded and binary responses and illustrate how the two response modes are represented by FRA. Besides, IFN- γ responses in two cancer cell lines were studied to demonstrate how mean responses may mask stimulation sensitivity in cancer cell populations, which can be rescued by FRA.

Discrimination between gradual and an all-or-none-type responses - synthetic toy example

In the deployed *in silico* model the same mean response results from different response distributions (unimodal vs bimodal) in cellular populations, which is adequately captured by FRA as presented in Supplementary Notes, Section 3.1, and Supplementary Fig. 13.

Discrimination between different modes of responses in cancer cell lines

In order to further demonstrate how FRA can uncover patterns masked by population averages, we performed IFN- γ dose-response experiment on two lung cancer cell lines: A549 and CALU1. We demonstrated that a comparison of cells' sensitivity to stimulation in terms of mean responses could be misleading. In terms of fractions of cells that exhibit different responses due to dose change CALU1 is more sensitive to IFN- γ than A549, whereas the mean response suggests the opposite, Supplementary Figure 14. We describe this phenomenon in detail in Supplementary Notes, Section 3.2.

FRA estimation accuracy

In order to examine how the estimation of FRA depends on the data size and data dimensionality, we have considered simple test models and compared estimation results to numerically computed, exact values, Supplementary Notes, Section 4. A biased estimation can be expected if the data size is small compared to data dimensionality. However, in the test model, for one-dimensional data, eight measurements per dose gave accurate estimates. Accurate estimation for 100-dimensional data required more than 250 measurements per dose, Supplementary Fig. 15a,b. Most of the current high-throughput technologies provide measurements of hundreds, rather than tens, of cells per sample. Therefore, FRA

should provide reliable analysis even for highly-dimensional data as long as hundreds of cells per sample are available.

3. Interpretation of FRA

We modified the algorithmic description to Fig. 2 to more explicitly reflect the biological interpretation of FRC: the cumulative fraction of cells with different responses. Besides, in Supplementary Table 1, we provide a summary of different interpretations of FRA. Also, Section 2.4 of Supplementary Notes utilizes the GPCR signaling data set to provide further guidance on the interpretation of FRA.

Reviewer #1

The authors present a “fractional response analysis” (FRA) framework to quantify responses to signals using a variant of Renyi min entropy, a quantity from information theory and an alternative to the more frequently used Shannon entropy. It is suggested that this formalism has advantages to conventional approaches especially when analyzing high-dimensional single-cell data. As biological applications, the authors performed and analyzed a CyTof experiment with PBMCs after stimulation with IFN- α , and a microscopic time-series experiment of U937 macrophage-like cells after stimulation with IFN- γ or IL-10. Finally, a published kinetic data set on TNF-stimulated fibroblasts was re-examined. As a mayor conclusion, the authors highlight applicability of the Weber-Fechner law to cellular responses to cytokines.

The FRA framework itself and the application of Renyi entropy to signal transduction in biology is a very interesting and timely idea. Indeed, quantitative methods for analysis of single-cell data are highly needed in the field. Moreover, the authors made an effort to provide fresh single-cell data from highly relevant biological model systems (cytokine-driven pSTAT activation in different cell lines), suitable for application of their new FRA approach. However, the downside is that this approach, using a new method with new data, also limits comparison to existing data sets and analysis methods. Therefore, I would recommend adding analysis of published data sets by FRA and/or providing a fair comparison of FRA to a set of other analysis methods on the new data set.

We have addressed this concern in point 1 of the general response.

In the presented workflow, this kind of analysis is limited to t-SNE plots and mean intensity plots shown in Fig. 1, which cannot really be compared to FRA (see below for details). In summary, before publication, the authors should definitely clarify and substantiate their claim in the abstract, that FRA “uncovers otherwise hidden patterns in single-cell data”.

We have rephrased our claim by changing “uncovers” into “quantifies”. We believe that the claim is substantiated with analysis of CyTOF data that shows all types having nearly identical behaviour in terms of fractions of cells with different responses. This observation cannot be made with other types of analysis. To be more clear why our observation is new, we have expanded our discussion by explaining how this observation is different from previous studies reporting Weber's law type of behavior in the cellular signaling systems. In short, we demonstrated that within heterogeneous populations of cells of a given type, and across types, a number of cells that exhibit a different response is proportional to the fold-change in the dose. We did not refer to a single signaling effector in a representative cell or population-average but to the state of the heterogeneous population described by multivariate data.

Major comments:

- The proposed mathematical framework based on Renyi entropy is the most interesting part of the study. However, a number of questions remain open, conceptually as well as regarding interpretation and (lack of) comparison to other measures of channel capacity. What is the advantage of using Renyi min entropy with respect to the more classic Shannon entropy? How does the proposed framework relate to other multiple-input information theory methods, such as those recently published by the same group (Jetka et al., PLoS Comp Biol 2019; Jetka et al., Nat Comm 2018)? Is it possible to give a more intuitive explanation of what FRA actually quantifies – in addition to the more algorithmic description given in Fig. 2?

The newly added Section 2 of Supplementary Notes contains a thorough comparison of Renyi min-information and the Shannon approaches, whereas subsection 2.2 explains the advantages of Renyi min-information.

In short, Renyi min-information is expressed in terms of fractions of cells with different response levels and, therefore, provides a clear and intuitive interpretation. In contrast, the interpretation of Shannon information for signaling data is problematic. Primarily, it implicitly assumes that cell-to-cell heterogeneity constitutes noise leading to information loss, which is largely not true. Besides, the interpretation of Shannon information is based on Shannon coding theorem, which assumes that coding and decoding are performed on long sequences of discrete symbols, as in electronic communication, which is not the case for cellular signaling.

- Technically, I am wondering whether the proposed FRC quantity fulfills some type of “associative law”. Meaning, if we have measurements at dose x_1 , x_2 , x_3 , I understand that $r(x_2) = r(x_1) + dr_{21}$, and $r(x_3) = r(x_2) + dr_{32}$ (Fig. 2C,D) where dr_{ij} is the area under the curve difference between the response distributions $r(x_i)$ and $r(x_j)$. Now, does it hold true that $r(x_3) = r(x_1) + dr_{31}$? If not, would that not have severe implications for interpretation, and even more so for comparison of different

data sets? In more general terms, it appears to me that FRA quantifies a specific experiment rather than a biological system – everything depends on the number and location of considered doses (or other measured signals). That does not mean FRA cannot reveal important insights, but such matters should be clearly discussed in the paper, and compared to other methods.

This is a very interesting point, which relates to the general properties of information quantification stemming from axiomatic definitions of Shannon and Renyi. In short, the “associative law” would hold only under continuous input, where “dr” is infinitesimal. FRC satisfies axioms expected from a reliable information measure, as proposed by A. Renyi, which guarantees that the measure does not violate any fundamental laws. Therefore, not obeying “associative law” is not a caveat *per se* but a necessity resulting from a need not to quantify redundant information.

Nonetheless, the reviewer is correct that FRA quantifies a specific experiment rather than a biological system. However, a similar critique could be made for many other useful measures, including Shannon information. If doses are too sparse or don’t cover the relevant dynamic range, the mean dose-response will not be representative of the system but only of the specific experiment.

Appreciating the value of this remark, we have added Section 4.2 to the Supplementary Notes, which shows how FRC depends on the number of doses used in an experiment.

- The shown “conventional analysis” in Fig. 1, and its relation to the proposed new framework, is enigmatic. P.3, “A single t-SNE plot represents responses in terms of one signaling effector only” – that is not necessarily true, and usually not the purpose of t-SNE plots: as shown in Fig. 1A, the main purpose of t-SNE visualization and related clustering methods is classification of single cells and not dose-response analysis. How could such an analysis be replaced by the proposed FRA method? From what is shown in Fig. 3, it rather seems that a cell type analysis method such as t-SNE is a prerequisite for FRA. Further down: “Typically, to obtain quantitative characteristics ... mean responses ... are plotted” – that is not true, in many standard flow cytometry analysis pipelines, the geometric mean index is used for quantification rather than mean intensity. It would be very interesting to see whether that quantity, and/ or a simple log-transformed intensity value, behaves similar to the FRA in a direct comparison to Fig. 3A.

This is a very adequate point. Indeed, t-sne plot is a prerequisite of a more detailed analysis, including FRA. Therefore, both approaches are complementary. We should have pointed to complementarity rather than to the opposition. We have introduced the necessary changes where suitable, including the first section of Results.

Mean fluorescence intensity (MFI), along with median and indeed geometric MFI, are the most prevalent representation of flow cytometry data. We have corrected the phrasing to be more precise and to account for the geometric mean.

FRC accounts simultaneously for several signaling effectors, in contrast to the arithmetic mean, geometric mean, or median. Therefore, a comparison of FRCs presented in Fig. 3a with geometric means is not directly possible. We have, however, plotted the geometric means in Supplementary Fig. 3a. Geometric mean is indeed more similar to FRC of individual signaling effector, Supplementary Fig. 5, than arithmetic means or medians, most likely due to the fact that univariate response distributions in CyTOF data are closer to log-normal than normal distributions. We have added suitable explanations to the text.

- In light of the research questions and claims raised in the introduction, the conclusions and the shown analyses of examples are rather underwhelming. In particular, it is suggested that “conventional dose-response curves do not capture the inherent complexity of single-cell high-throughput data” (p.2), with reference to phenomena such as cell-to-cell heterogeneity or signals that change over time. In contrast to that task list, in the results part, the number of parallel responses analyzed does not exceed 5 (pSTAT1,3,4,5,6), and the paper is largely concerned with analyzing single time points (Fig. 1-3 and Fig. 4A,B,D,E) - only Fig. 4 C,F is concerned with time-series analysis, with very limited conclusions. Again, would it be possible to add applications of FRA to more complex, published data sets, and compare the conclusions to those from the original work?

Our CyTOF data incorporate 6 effectors at a single time-point in 5 different cell-types, whereas NF- κ B data the single effector at 21 time-points. Therefore, the data are multivariate, and FRC demonstrates how multivariate response can be aggregated in an individual response curve.

We agree with the reviewer that, indeed, analysis of a data set that accounts for multiple signaling effectors at multiple time-points could show the full potential of our approach. Experimental methodologies to perform such experiments are however, very limited. CyTOF, and flow cytometry, allows measurements of multiple signaling effectors at a single time-point. Life microscopy imaging, on the other hand, requires a construction of a fluorescent reporter for each signaling effector, which limits the number of signaling effectors that can be quantified in the same cell. We are not aware of a publicly available data set that describes temporally resolved dose-responses of individual cells in terms of multiple signaling effectors measured in the same cell. If the reviewer is aware of a data set with multiple signaling effectors measured over-time in individual cells for several doses, we would be very interested in performing its analysis within our framework. The computational component of our approach is based on logistic regression, which is known to work

well for highly dimensional data. Therefore, the number of covariates is not a limiting factor for the applicability of FRA.

It is likely that technologies enabling measurement of multiple signaling effectors over-time in the same cell will be developed in the future. In such a case, FRA will be readily deployable.

- The main biological discovery, applicability of Weber's law to cytokine signaling, is interesting but not that new or surprising as suggested by the authors. The discovery of fold-change detection in eukaryotic signaling goes back (at least) to a series of papers on Wnt signaling published in 2009 in Mol Cell (Goentoro and Kirschner; Goentoro ... Alon; James Ferrell). In contrast to the claim of the authors that "a mechanistic explanation ... remains to be determined", those papers already proposed an incoherent feed-forward loop motif as a plausible mechanism. More recent work described fold-change detection or Weber's law in context of NFkB (Lee ... Gaudet, Mol Cell 2014) and Ca(2+) signaling (Thurley et al., Sci Signal 2014), to name only a few. Further, it remains unclear how much the described new information-theoretic framework contributed to that major finding of fold-change detection. Would the methods used in the papers mentioned above, which also analyze single-cell data, fail to discover fold-change detection in the data sets analyzed here?

The discussion has been rewritten to unambiguously explain the differences between our observation and previous works.

Our observation is considerably different. We demonstrated that within a heterogeneous population of cells of a given type and across types, a number of cells that exhibit a different response is proportional to the fold-change in the dose. In contrast to previous works, we did not refer to a single signaling effector in a representative cell or population-average but to the state of the heterogeneous population described by multivariate data. Ultimate outcomes of multicellular processes like immunity are not determined by individual cells alone or population averages but by the heterogeneous collective. By accounting for cell-to-cell heterogeneity, we showed that the distribution of the collective, which encodes stimulation level in multicellular systems, shifts with the fold-changes of the dose.

- P. 6, "for IFN-g, distributions of responses shifted gradually ... For IL-10 the distributions flattened ... similar to a binary system". From my understanding, this is one of the most interesting observations in the manuscript. Could FRA be used in more general terms to distinguish a gradual from an all-or-none-type of response, or even better to quantify the level of gradual response that can be achieved by a certain signaling pathway? If that is the case, it would be very helpful to compare

FRA to “conventional methods” in this regard, such as fitting of the dose-response (possibly in terms of geometric mean index) to a Hill curve.

This was indeed a very useful suggestion. We explained how it was addressed in point 2 of the general response. In particular, the remark inspired Section 3 of Supplementary Notes and Supplementary Figure 13.

Minor comments:

- It would be advisable to avoid wordings such as “typically” or “conventional methods” without any reference.

Text alterations that introduce a more careful use of these words clearly improved the quality of the text and the precision of communication.

- The t-SNE analysis and cell clustering performed in Fig. 1, and used for further analysis in Fig. 3, should be described in much more detail in the methods section, preferably accompanied by giving access to the source-code for the analysis.

We amended the github repository with an R-script with references to the specific libraries used for computations so that the plot can be reproduced by the interested researchers. The data are deposited in the repository as well.

- I did not yet evaluate the provided source code for calculating FRA, which should be done at a later stage of the review process if applicable.

Reviewer #2

In “Fractional response analysis reveals logarithmic cytokine responses in cellular populations,” the authors present a novel tool, namely the fractional response analysis (FRA), to quantify and interpret changes in the heterogeneity of single-cell signaling responses after cell stimulation, and for responses obtained by high-throughput techniques. The scientific question is timely and relevant, as the quantification and interpretation of single-cell responses is a fundamental concern of many investigations in cellular biology. The paper is also interesting in that the authors present high-quality experimental data for multiple different combinations of cell lines and conditions, and they analyze these data using their new metric of fraction response.

The proposed method may potentially reveal new insights into single-cell responses, but as the paper is current written, the actual benefits and limitations of the fractional response analysis are not adequately described or tested. Without a more rigorous investigation of the metric and its ability to discriminate between different models or

different biological circumstances, the importance of this new tool is not clear. More detailed comments are provided as follows:

The above was addressed in the general response.

1) The authors introduce in the main text three main difficulties to interpret single-cell signaling data: 1) cell-to-cell heterogeneity, 2) complex responses given the signaling network, and 3) transient responses. The paper falls somewhat short of addressing these issues in different ways.

1a. Regarding heterogeneity: the authors discuss the limitation of conventional dose-response curves commonly used to report single-cell response analysis. The advantage of the use of FRA vs dose-response curves are clear, but this in itself is not an original observation. One of our main concerns is that the authors don't adequately compare their method existing work that has also sought to integrate stochastic models and single-cell data to improve the quantitative characterization of single-cell responses. A very large number of groups have used noise as an additional feature to discriminate between different cell populations with similar means (see for example the work from: Nicolas Buchler, Hana El Samad, Ramon Grima, Srividya Iyer-Biswas, Mustafa Khammash, Heinz Koepl, Thomas Lipniaki, Andrew Mugler, Brian Munsky, Rob Phillips, Arjun Raj, Jakub Reuss, Abhyudai Singh, Eduardo Sontag, and Alexander van Oudenaarden). While it is understandable that no paper could possibly cite all relevant work on this well-studied topic (and we do recognize that other relevant groups are cited but not discussed in any detail), we would expect to see at least some comparison of the FRA to other metrics to quantify differences in cellular variability from one population to another.

The above concern was addressed in point 1 of the general response. Besides, we have complemented our reference list with two comprehensive references discussing issues problems of cell-to-cell heterogeneity quantification:

-Eling, Nils, Michael D. Morgan, and John C. Marioni. "Challenges in measuring and understanding biological noise." *Nature Reviews Genetics* 20.9 (2019): 536-548.

-Symmons, Orsolya, and Arjun Raj. "What's luck got to do with it: single cells, multiple fates, and biological nondeterminism." *Molecular cell* 62.5 (2016): 788-802.

1b. Regarding complexity: the authors state that the proposed FRA is amenable to high-content data (e.g., data in which many different transcription factors or reporters are measured simultaneously), and this claim is the main focus of the final paragraph of the discussion. However, the paper itself focuses only on one-dimensional data (i.e., one distribution at a time). The FRC is very easy to understand in 1D when dealing with one feature at a time, but it is not clear how well the method scales with larger dimensions of data. Would the FRC be defined separately and summed over each marginal (as seems to be the case in the current manuscript), or over the full

N-dimensional joint distribution. If the former, then how would one detect or account for dependencies in the data in which one feature is strongly correlated with another? If the latter, then how would the method behave as the number of bins grows exponentially with the number of species and the sampling of these bins becomes extremely sparse?

In fact, we focus on multivariate data. Our CyTOF data incorporate 6 effectors at a single time-point in 5 different cell-types, whereas NF- κ B data involve the single effector measured at 21 time-points. Such output dimensionality is considerably higher compared to what could be analyzed with previous methods derived from information theory.

In addition, the computational component of our approach is based on logistic regression and does not require binning as many other strategies originating information theory. Logistic regression is known to work well with high-dimensional data. Therefore, dimensionality is not a limiting factor for FRA. In fact, logistic regression not only overcomes binning but enables breaking an important computational barrier related to data dimensionality, which we believe to be a relevant, innovative component of our approach.

The examples with one-dimensional data, IFN- γ and IL-10 responses in U937 cell lines, were presented to show how FRA represents qualitatively different response types, which would not be visible for multivariate scenarios.

We have altered the text, where possible, to indicate more clearly that we analyze multivariate data.

1c. It was stated that FRA could be performed for dynamic measurements. However, the examples seem only show FRA performed on fixed-time population snapshots per dose. Do the authors believe that FRA could reveal patterns in the temporal evolution of the response as well? If so, how would that be done?

We have altered the text where possible to indicate more clearly that we analyzed time-series data for NF- κ B responses.

NF- κ B example examines high-dimensional life imaging data. The response of each cell is represented by a 21-dimensional vector. Quantification of how these 21-dimensional distributions shift with increasing dose revealed sensitivity to fold-changes, which was in line with other analyzed cytokine responses.

The dimensionality of data is not a limiting factor for the applicability of FRA. In an ideal scenario, the applicability of FRA should be demonstrated with data involving multiple signaling effectors measured over-time in single-cells for several doses. Acquiring such a data set with current technologies is, however, problematic, and we

are not aware of such a large-scale data set to be publically available. Nevertheless, it is likely that future technologies will enable the measurement of multiple signaling effectors over-time in the same cell in a straightforward way. In such cases, FRA will be readily deployable.

2) It seems that the FRA would be sensitive to the choice of binning strategies. We would expect that if bins are too coarse, then the differences between distributions would become negligible and the FRA would not work. If the bins are too fine, then finite sample sizes will make any two probability densities look unique from one another. This issue is particularly important when dealing with multiple features. For example suppose that the measurements include 10 features, each of which can take on values in 5 separate bins. This would suggest up to 5^{10} different bins – how reproducible would the FRC results be under finite amounts of data? How sensitive are the FRA results to specific choices of bins?

As explained when addressing previous points, no binning is involved thanks to the use of logistic regression. We have added additional clarifications where possible to highlight how using logistic regression overcame binning.

3) The FRA offers a simple method to interpret single-cell signaling responses, but at the same time, it is not very clear what is the resolution power of this methodology.

3a. For example, for pStat1 in Fig. 3-a, all the obtained results look very similar for the different cell types, even when the distributions look quite different among the cell types (Fig. 1D). For pStat5 in Fig. S5, the FRA does reveal a difference between NK cells and the rest (this difference is also obvious in Fig 1D).

Indeed, different STATs exhibit different response distributions in different cell types. Yet, FRA yields similar results. Both, FRC and cell-to-cell heterogeneity structure are similar across cell types. In our view, this is an unexpected result of considerable biological significance: despite differences in mean responses and degrees of cell-to-cell heterogeneity, all cell types follow the same pattern, i.e., Weber-Fechner law. The fraction of cells that exhibit different responses is proportional to the relative change in the dose.

We have added a more detailed description of this phenomenon in the Discussion and the Results.

3b. The paper seems to focus on discussing the similarity of the FRA for different cells and different conditions, but it is not clear what is the utility of a metric that gives the same result for most situations. Rather, it would be better to focus on when the FRA reveals differences that might have been missed using other methods.

This was indeed a very useful suggestion. We explained how it was addressed in point 2 of the general response. In particular, the remark inspired Section 3 of Supplementary Notes and Supplementary Figure 13.

4) Overall, the paper as written does not provide a convincing argument for if/when the FRA should be expected to provide discriminatory insight between different cells or different response types.

4a. The paper would be substantially improved with a more rigorous investigation of the FRA metric in a controlled theoretical or computational framework. Performing the analysis on model-generated data (i.e., with a known ground truth) would enable to the authors to better probe what kinds of features can the FRA reveal, and what are the limitations of the analysis. For example, the authors could employ several toy models of stochastic gene regulation (maybe even just using simple bursting gene expression models with different regulatory mechanisms or in different parameter regimes), and demonstrate clearly when the FRA is able to discriminate between different models and when it is not.

Again, this was a very valuable comment which led us to demonstrating advantages more broadly, particularly in Sections 2 and 3 of Supplementary Notes.

4b. With simulated data, the authors could probe the effectiveness of the FRA to discriminate different phenotypes and see how that depends on (i) different types of binning, (ii) different transformations of data (e.g., log or linear space), (iii) different sample sizes, (iv) different amounts or types (e.g., additive, multiplicative) of experimental measurement noise, (v) different ranges or resolutions of stimuli, (vi) different shapes of distributions (unimodal, bimodal), (vii) different numbers and dependencies of feature measurements, etc. While not all of these would need to be addressed, the authors should at a minimum show some examples of effects that are well treated using FRA and ideally they could also discuss some that are limitations of FRA.

This is indeed an important point that we have neglected in the previous version of the manuscript, perhaps due to our previous experiences with logistic regression, which works surprisingly well in a wide range of examples. The critique inspired us to discuss limitations of FRA in Section 4 of the Supplementary Notes.

5) In Figures 3A, 4 D-F, and S5, the color bands are confusing and do not provide much insight as described. Are these just showing the same fraction information as shown in Fig. 3B? If so, the pie charts are much easier to interpret. Either way, the caption needs to include a better description of the figure.

Indeed, pie charts are much easier to read. The bands show the same information as the off-diagonal elements of the pie charts. Therefore, the presented information

is largely redundant. Individual colors are difficult to read, but the overall height of the bar is well visible. The overall height of the bands quantifies to what extent cellular population stimulated with a specific dose exhibits responses overlapping with all other distributions providing insight on the extent to which a given distribution is distinct. It is a measure of signaling variability in relation to other distributions, with a narrow band indicating low variability and high band large variability. Variability is typically denoted as an error bar. Therefore, we believe that plotting bands around the curve, in addition to the pie charts, is a sensible solution.

Having the appreciation for this remark, as it was also our concern when designing the FRA plots, we added an option to turn off plotting of the bands in the associate software R-package, so that future users can decide for themselves what is the most useful and transparent presentation form.

6) On page 6, the sentence, “On the other hand, TNF- α responses are given as time-series, therefore, their characteristics cannot be directly observed.” was not clear.

Why does time series data prevent observation of the data characteristics?

The sentence was indeed difficult to comprehend, and it is now clarified. What we meant was the following: the time-series corresponding to single-cells stimulated with a given dose constitute multivariate distributions. Therefore, overlaps between distributions corresponding to different doses, Fig. 4c, are not easily visible and cannot be quantified in a straightforward way. FRA, on the other hand, enables both the quantification and visualization of the overlaps.

REVIEWERS' COMMENTS

Reviewer #1 (Remarks to the Author):

The authors did a very good job in review, the manuscript is now much more balanced and comprehensible. The discussion of fold-change detection is much more convincing now. Also, the additional analyses and discussions presented in the supplement, regarding comparison of FRA to other analysis methods and graded vs binary response, substantially strengthen the paper. I find it very interesting that it was possible to generate a graded and a binary version of a model with the same dose-response, which could be discriminated by FRA (Suppl. Fig. 13) - that could be highlighted even more in the main paper (just a suggestion).

However, I am still missing two points in that last discussion:

- Graded vs. binary response is now extensively analysed in the supplement for an in silico model and for an additional data set, but for the original data presented in the main paper (Fig.4 d-e), I cannot find the corresponding MFI or geo mean curves for comparison.
- The whole analysis remains on a rather qualitative level. Would it be possible to quantify the whole discussion about present and missing bands etc. in Supplementary Section 3, to give a more precise definition of graded or binary response based on FRA plot? In any case, it would help to add the Hill coefficients in Suppl. Fig. 14 a-b and in the corresponding data for Fig. 4 d-e).

Once these few points are clarified, in my opinion the paper is ready for publication.

Reviewer #2 (Remarks to the Author):

In the revision of "Fractional response, analysis reveals logarithmic cytokine responses in cellular populations," the authors have addressed our previous comments in that i) the supplementary information now contains a more detailed mathematical description of the FRA, ii) the text now describes the implementation of FRA and its applications, iii) the document now includes a supplementary section (SI 4) presenting some of the limitations of the proposed metric, and iv) the authors extended their analyses to include published data sets and the comparison with existing metrics to demonstrate the application and scope of the FRA. In general, we feel that the document is complete, and that the authors appropriately describe their experiments, figures properly demonstrate their results, and figure legends are complete.

Overall, we believe that FRA is a strong concept in its simplicity and ease of implementation – this makes it of broad interest to the readership of Nat Comms. The fact that the FRA must be estimated from finite data remains a limitation that is not yet fully understood. More mathematical analysis, improved estimators, and more thorough uncertainty quantification will be needed before FRA could be used reliably on complex datasets. But to be fair to the authors, we do not expect every issue to be fully addressed in the foundational work for a new method. For those reasons, we have no further major critiques, and only minor or cosmetic amendments are needed on the current text:

1) The paper is still not as clearly written as it could have been. The FRC is a very straightforward approach, and although the details are included in the manuscript methods and SI, they are difficult to find. The reader should not be forced to work so hard to understand how the method is to be applied to multivariate data. Some simple edits could make this clearer to the reader as follows:

Page 4 (near the bottom after the statement "The FRC can be universally calculated for any type of signaling data, i.e., arbitrary number of signaling effectors, time points of measurements, doses, or other experimentally varied parameters.") -- Please remind the reader know that $P(y|x)$ does not need to be densely sampled, but it can be approximately estimated from finite data via probability density estimators and refer to the methods text where this is described.

Page 5 -- in discussing the FRA, please refer to the methods section and mention the proposed use of logistic regression. This allows for a treatment that is appropriate for continuous-valued multivariate data.

Page 2 -- Please give more details about the types and characteristics of the "26 phenotypic markers" that were measured.

2) Limitations of the FRA metric are not discussed sufficiently in the main text. The authors provide new numerical tests in the Supplementary Section 4 and Sup Fig 15 ("Caveats of FRA"). Overall, these tests (done on a simple model) suggest that the statistical estimation of the FRC becomes more accurate when the number of samples increases and is greater than the dimension of the response/output. However, this estimator appears to be biased when the sample size is about the same or slightly more significant than the output dimension. The error bars seem to underestimate the true uncertainty when $d = 10$ or 100 . In this case they seem to suggest perfect discrimination between the two inputs. The FRA also depends on the doses chosen to generate the data, which the authors discuss in Supplementary Section 4.2. This indicates that the properties of its statistical estimator from finite datasets are not fully understood. There needs to be references to these limitations in the main text. Specifically, we would like to see a qualitative discussion (if not necessarily a rigorous quantitative description) of how the number of required samples needed to estimate the FRC depend on: (i) the number of features collected in the data, (ii) the specific probability density estimators or regression models, (iii) the underlying dependency relationships between inputs and the downstream multivariate responses. The manuscript should provide some words of caution in the discussion section about FRA's limitations when applied to finite dataset (e.g., the method may be appropriate for high-throughput-moderate-content CYTOF data, but not for higher-content-lower-throughput approaches like single-cell sequencing or spatial transcriptomic analyses using multiplexed FISH).

We thank both reviewers for their efforts to improve our manuscript, which allowed us to identify our blindspot and lack of balance in certain arguments. We have altered the manuscript along the suggested lines and believe that the manuscript substantially benefited from reviewers' comments.

Below we provide a point-by-point response, where reviewers' comments are in black, and our response in blue.

REVIEWERS' COMMENTS

Reviewer #1 (Remarks to the Author):

The authors did a very good job in review, the manuscript is now much more balanced and comprehensible. The discussion of fold-change detection is much more convincing now. Also, the additional analyses and discussions presented in the supplement, regarding comparison of FRA to other analysis methods and graded vs binary response, substantially strengthen the paper. I find it very interesting that it was possible to generate a graded and a binary version of a model with the same dose-response, which could be discriminated by FRA (Suppl. Fig. 13) - that could be highlighted even more in the main paper (just a suggestion).

We highlighted the reference to the Supp. Fig 13 (now Supp. Fig 15) in the main text.

However, I am still missing two points in that last discussion:

- Graded vs. binary response is now extensively analysed in the supplement for an in silico model and for an additional data set, but for the original data presented in the main paper (Fig.4 d-e), I cannot find the corresponding MFI or geo mean curves for comparison.

We prepared a new supplementary figure that compares FRC and geometric mean (mean of log-data). The figure is shown as Supp. Fig 12.

- The whole analysis remains on a rather qualitative level. Would it be possible to quantify the whole discussion about present and missing bands etc. in Supplementary Section 3, to give a more precise definition of graded or binary response based on FRA plot?

The discussion of Supplementary Section 3 was amended to explain in more detail how the graded and binary responses can be discriminated based on the FRA plot, in particular from present and missing bands.

In any case, it would help to add the Hill coefficients in Suppl. Fig. 14 a-b and in the corresponding data for Fig. 4 d-e).

We are not sure how to address this remark, as no Hill coefficients were involved in these plots. In addition, we are not convinced what benefits fitting the Hill equation to the obtained curves could bring. Therefore, we decided not to change the above figures.

Once these few points are clarified, in my opinion the paper is ready for publication.

Reviewer #2 (Remarks to the Author):

In the revision of "Fractional response, analysis reveals logarithmic cytokine responses in cellular populations," the authors have addressed our previous comments in that i) the supplementary information now contains a more detailed mathematical description of the FRA, ii) the text now describes the implementation of FRA and its applications, iii) the document now includes a supplementary section (SI 4) presenting some of the limitations of the proposed metric, and iv) the authors extended their analyses to include published data sets and the comparison with existing metrics to demonstrate the application and scope of the FRA. In general, we feel that the document is complete, and that the authors appropriately describe their experiments, figures properly demonstrate their results, and figure legends are complete.

Overall, we believe that FRA is a strong concept in its simplicity and ease of implementation – this makes it of broad interest to the readership of Nat Comms. The fact that the FRA must be estimated from finite data remains a limitation that is not yet fully understood. More mathematical analysis, improved estimators, and more thorough uncertainty quantification will be needed before FRA could be used reliably on complex datasets. But to be fair to the authors, we do not expect every issue to be fully addressed in the foundational work for a new method. For those reasons, we have no further major critiques, and only minor or cosmetic amendments are needed on the current text:

1) The paper is still not as clearly written as it could have been. The FRC is a very straightforward approach, and although the details are included in the manuscript methods and SI, they are difficult to find. The reader should not be forced to work so hard to understand how the method is to be applied to multivariate data. Some simple edits could make this clearer to the reader as follows:

Page 4 (near the bottom after the statement "The FRC can be universally calculated for any type of signaling data, i.e., arbitrary number of signaling effectors, time points of measurements, doses, or other experimentally varied parameters.") -- Please remind the reader know that $P(y|x)$ does not need to be densely sampled, but it can be approximately estimated from finite data via probability density estimators and refer to the methods text where this is described.

Explanation was added.

Page 5 -- in discussing the FRA, please refer to the methods section and mention the proposed use of logistic regression. This allows for a treatment that is appropriate for continuous-valued multivariate data.

Additional references to Methods as well as further clarifications were added.

Page 2 -- Please give more details about the types and characteristics of the "26 phenotypic markers" that were measured.

We exemplified in the main text the type of phenotypic markers used and added a reference to the Supplementary Table 1. Also, in the Supplementary Table 1 we marked which of the used antibodies are against phenotypic markers.

2) Limitations of the FRA metric are not discussed sufficiently in the main text. The authors provide new numerical tests in the Supplementary Section 4 and Sup Fig 15 ("Caveats of FRA"). Overall, these tests (done on a simple model) suggest that the statistical estimation of the FRC becomes more accurate when the number of samples increases and is greater than the dimension of the response/output. However, this estimator appears to be biased when the sample size is about the same or slightly more significant than the output dimension. The error bars seem to underestimate the true uncertainty when $d = 10$ or 100 . In this case they seem to suggest perfect discrimination between the two inputs. The FRA also depends on the doses chosen to generate the data, which the authors discuss in Supplementary Section 4.2. This indicates that the properties of its statistical estimator from finite datasets are not fully understood. There needs to be references to these limitations in the main text. Specifically, we would like to see a qualitative discussion (if not necessarily a rigorous quantitative description) of how the number of required samples needed to estimate the FRC depend on: (i) the number of features collected in the data, (ii) the specific probability density estimators or regression models, (iii) the underlying dependency relationships between inputs and the downstream multivariate responses. The manuscript should provide some words of caution in the discussion section about FRA's limitations when applied to finite dataset (e.g., the method may be appropriate for high-throughput-moderate-content CYTOF data, but not for higher-content-lower-throughput approaches like single-cell sequencing or spatial transcriptomic analyses using multiplexed FISH).

These are indeed essential points to fully explore limitations and strengths of FRA. In the revised manuscript, we elaborated more on the implication of our test model of Supplementary Section 4.2, and added an additional panel to the Supplementary Fig. 17, which shows how the required sample size depends on the dimensionality of the data. Also, in the main paper, we added additional references to the caveats of FRA. In the amendments, we intended to highlight that accurate estimation requires the cell number per dose to considerably exceed the number of measured signaling effectors. However, given the novelty of the method, a comprehensive exploration of the above issues would, in our view, deserve a standalone paper.